# Public attitudes toward medical waste: Experiences from 141 countries

**Zhipeng Bai** *, **Xi Liu, Wenbao Ma**

Department of Philosophy, Xian Jiaotong University, Xian, Shaanxi, China

* baizhipeng68@163.com

## Abstract

### Background

Medical Waste (MW), conceptualized as waste generated in the diagnosis, treatment, or immunization of human beings or animals, posing massive threat to public health. Environment-friendly public attitudes promotes the shaping of pro-environmental behavior. However, the public attitudes of MW and the potential determinants remained scarce. The present study aims to reveal globally public attitudes towards MW and captured the determinants.

### Methods

We integrated the crawler technology with sentiment analysis to captured the public attitudes toward MW across 141 specific countries from 3,789,764 related tweets. Multiple cross-national databases were integrated to assess characteristics including risk, resistance, environment, and development. The spatial regression model was taken to counterbalance the potential statistical bias.

### Results

Overall, the global public attitudes towards MW were positive, and varied significantly across countries. Resilience (β = 0.78, SD = 0.14, P < 0.01) and development (β = 1.66, SD = 0.13, P < 0.01) posed positive influence on public attitudes towards MW, meanwhile, risk (β = -0.1, SD = 0.12, P > 0.05) and environment (β = 0.09, SD = 0.09, P > 0.05) were irrelated to the shaping of positive MW public attitudes. Several positive moderating influences was also captured. Additionally, the cross-national disparities of the determiants were also captured, more specific, public attitudes towards MW in extremely poor areas were more likely to be negatively affected by risks, resilience and development.

### Conclusions

This study focused mainly on the public attitudes as well as captured the potential determinants. Public attitudes towards MW were generally positive, but there were large cross-national disparities. Stakeholders would need to designate targeted strategies to enhance public satisfaction with MW management.

**Data Availability Statement:** All relevant data are within the paper and its Supporting Information files. Additional data can be requested by contacting the corresponding author.

**Funding:** The authors received no specific funding for this work.

**Competing interests:** The authors have declared that no competing interests exist.

## Section 1: Introduction

Medical Waste (MW), defined by the World Health Organization (WHO) as the waste generated in the diagnosis, treatment, or immunization of human beings or animals [1]. The established study indicates that the production of MW has risen 14 times from 2010 to 2023, which posing massive threat to public health [2]. This exponential global rise in MW is derived from three perspectives as: Firstly, the world is experiencing rapid ageing, leading to the overuse of medical devices to manage age-related illnesses. Secondly, the healthcare industry is shifting from multi-use medical devices towards safer, single-use medical devices, which adds to the production of MW. Thirdly, external risks such as widespread epidemics (like Corona Virus Disease 2019, COVID-19) further raising the concerns about the health of the global population and generating numerous Personal Protective Equipment (PPEs).

The MW brings threefold hazards. Firstly, improper handling and disposal of MW pose a significant risk of infection, and deduce severe health risks. Secondly, the multiple using of MW causes cross-infection of diseases such as HIV, hepatitis B, and hepatitis C. Thirdly, the combustion of MW results in the countless polychlorinated dibenzopdioxins and polychlorinated dibenzofurans, shaping detrimental impact on human health and the environment [3, 4]. Additionally, another significant challenge of MW disposal is the economic burden. Properly disposal MW is economically difficult for the vast majority of countries. More specific, in the United States, scientifc disposing of infectious MW costs $0.79 per kilogram, equally to 5.6 times compared to it's unscientific counterpart [5]. More critically, there are significant differences between developed countries and developing or lagging countries in terms of the amount of MW emissions, the scientific validity of MW disposal, and the amount of pollution generated by MW. The above scenario reveal that the difference of public attitudes towards MW between countries at different development situation would result in significant national disparities. Accordingly, analyzing the science management MW framework should be posed at high academic priority.

Public attitudes toward MW is a crucial social determinant which influences MW management [6–9]. Environmental behaviorist have argued that pro-environmental public attitudes and perceptions contribute to the shaping of public environmental behaviors, and the above academic view has been widely proved in recycled water [10], air pollution [11], biodiversity [12], climate change [13], soil contamination [14]. Therefore, it is meaningful to shape a positive public attitude towards MW [6]. MW originates from personal daily life, including discarded masks, test papers, syringes, that means if without shaping a positive public attitude toward MW, these contaminants will be disposed of indiscriminately, and causing severe health risks [7]. Unfortunately, there is no established evidence on the public attitudes toward MW.

Therefore, our study use a combination of crawlers and sentiment analysis to collect 3,789,764 tweets from 141 countries, to reveal the cross-national heterogeneity of public attitudes towards MW, meanwhile, and captured the potential determinants of MW public attitudes cross-national heterogeneity. The three main aims of the present study are:

1. Describing the global geographic distribution of the public attitudes toward MW.

2. Discussing the cross-national heterogeneity of public attitudes towards MW.

3. Identifying whether there are significant differences in MW public attitudes between developed and developing or lagging countries.

4. Identifying the potential determinants related to the public attitudes toward MW.

This paper is organized as follows. Section 2 gives an overview of the MW related studies. Section 3 presents our theoretical framework and hypothesis. The methodological approach is explained in Section 4. Section 5 presents the main results, after which conclusion and discussion are shown in section 6.

## Section 2: Literature review

### 2.1. Medical waste

The announcement of the "U.S. Medical Waste Tracking Act of 1988" marked the first official definition of MW as "any solid waste generated in the diagnosis, treatment, or immunization of humans or animals, in research related thereto, or in manufacturing or testing biologicals" [15]. This definition was subsequently amended and widely adopted by the World Health Organization (WHO), which broadened the definition of MW to include not only solid waste, but also liquid and gaseous waste generated during diagnostic, therapeutic, or immunization procedures in humans or animals [16].

In a academic view, the meaning of MW was widely related to four specific terminologies, which were commonly interchanged with each other [17]. These included hospital waste, medical waste, regulated medical waste, and infectious medical waste [1]. The present study took the last on as the MW definition.

Overall, MW was divided into two categories: non-hazardous MW and hazardous MW. The former denoted the household medical waste or semi-household medical waste, which was harmless. The latter was the waste generated in hospitals, including dental waste, waste from medical laboratories and blood waste, which related to infectious and harmful [18]. WHO made a more detailed classification of MW, including "infection waste", "sharps", "chemicals", "pharmaceuticals" and "radioactive waste" [19]. The above scenario shown that MW was not only generated in medical facilities such as hospitals, dental clinics, laboratories, blood transfusion centers, but also derivde from individual households.

### 2.2. Medical waste management

The management of MW followed two main typical models: the government-led MW management model and the hospital-led MW management model. For one thing, the government-led MW management viewed MW as a controllable social behavior and therefore determined by public policy, attitudes and force. Accordingly, MW was a function of the counterbalance of competition and cooperation between different social stakeholders, Especially during the emergency period of the major pandemic like COVID-19. A cross-national empirical analysis showed that the government-led MW management model promote the MW scientific management, especially in several "taboo" fields [20]. Furthermore, most of the developing or lagging countries chose the government-led MW management, such as Nigeria, India, Angola, etc [8, 21, 22]. Taken together, government-led MW management shared the following three advantages convenience, efficiency, strictness, and suffered from several disadvantages monopoly, unscientific, rigid.

For another, hospital-led MW management was related to the MW management model as that hospitals possessed absolute authority over the generation, transportation, and disposal of MW. Several developed countries, such as the United States, Singapore, and South Korea, were inline with this model [4–6]. Furthermore, hospital-led MW management was consistent with the spirit of "Market Rational Thinking", which consider the MW management as the counterbalance between supply and demand [23, 24]. Accordingly, hospital-led MW management shared the advantages of economic, accessible, rational and the disadvantages of unfairness, excessive competition, profit-seeking.

Another concern of this study was the way MW was disposed of in different countries. There were two main MW disposal methods, a unscientific one, which was incineration and landfill. Incineration would result in the emission of large amounts of polychlorinated dibenzo-p-dioxins (dioxins) and polychlorinated dibenzo-p-dioxins (furans) and eventually pollute the air, and landfill also contain large amounts of toxic substances that would pollute the soil and water [25, 26]. These treatment methods were mainly found in developing or lagging countries such as Africa, South America and South East Asia [25, 26]. The other was the scientific approach, which consisted of chemical disinfection and steam sterilisation. They were both heat-based, safe and efficient treatment processes and were the most popular methods of MW disposal [27, 28]. However, due to their economic viability and the availability of treated wastes, their use was still limited to very few countries, such as developed countries in Europe, the United States and Japan [27, 28].

## 2.3 MW determinants

The established researches captured the following four potential macro determinants:

1. Risk. Conceptualized as the external pressure of MW [2, 5, 9]. This concept was also defined as "environmental risk perception" or "environmental perception" environmental science, and was proved as an important determinant in motivating individual pro-environmental behavior [5]. Several representative analysis revealed that risk contributed significantly to shaping cross-national heterogeneity in MW disposal and finally influence the MW management [6]. Meanwhile, other related researches revealed that the correlation between risk and scientific MW management was not consistent, and those relatively advantaged and disadvantage governments could benefit more from risk [29].

2. Resilience. Resilience referred to the ability to disposal excessive medical contaminants, as well as the accessibility and quality of healthcare [1]. In established related researches, communities with higher resilience were more likely to generate public pro-environmental behaviors and shaping scientific MW management, this scenario was captured in in both developed (USA, Japan, Korea) and developing countries (China, India), simultaneously [30, 31].

3. Environment. Environment in the present study denoted the natural environment, and was defined as the physical and biological systems related to the whole process of MW disposal and management [32]. Established literatures confirmed that living in a low-pollution environment promoting the shaping of pro-environmental behaviors such as waste sorting or water recycling for individuals [33, 34]. This scenario was attributed to that the low-pollution regions were equipped with more advanced MW disposal equipments [29].

4. Development. The concept of development encompassed the status of social conditions for individuals and communities. Environmental researchers assumed that a given country shared more scientific MW management under developed social stage [35, 36]. Meanwhile, the UN Sustainable Development Planning, 12 of the 17 quality development goals (assessing a given country's situation of development) set out the expectation of addressing the sustainability of MW [1, 2].

## 2.4. Research gap

We systematically reviewed the literatures on MW definition, management and determinants, and revealed the following academic scenario: MW derived not solely in hospitals, but in every household, meanwhile, the two available heterogeneous MW management models suffered

from different shortcomings, additionally, risk, resilience, environment and development influenced the MW. But there was no integrated framework for discussing the above issues, especially with regard to public attitudes towards MW. The present study aimed to narrowing this academic gap.

## Section 3: Methodology

### 3.1. Data collection

Twitter, the most representative social media worldwide, was the data source for the present study. We used the crawler technology to mining the data from twitter. Crawler techniques were particularly well suited to analyze public attitude towards MW, because Twitter provided the opportunity for the public to anonymously express their attitudes. We used URL-based crawler technology, which was proven to be a effective data acquisition way in academic analysis. Finally, we obtained 3,789,764 informative tweets from 141 countries about MW. Data were accessible and the data in S2 Table of S1 Appendix collection and analysis was fully compliant with the terms and conditions of the data source.

### 3.2. Sentiment analysis

Sentiment analysis was a natural language processing technique, which focused on analyzing public opinion and attitudes upon a given topics, and determined whether the sentiments was positive, neutral or negative. We used the unsupervised machine learning sentiments analysis developed by the National Research Center Canada (NRC). The NRC covered abundant words and their relationship to eight emotions, including anger, fear, sadness, joy, disgust, anticipation, trust and surprise. The first four of these were defined as negative emotions and the last four were defined as positive emotions.

All the steps of crawling and sentiment analysis were shown in Fig 1.

### 3.3. Variable selection

**3.3.1. Dependent variable.** The dependent variable we focused on was the Medical Waste tweets Positive-proportion (MWP), denoted whether the public attitude toward MW in a given country was positive or negative. We used a combination of crawling techniques and sentiment analysis to obtain the percentage of positive tweets about "medical waste" in 141 countries, which was a continuous variable and ranged from 63.1% to 74.4%. The more information of MWP was shown in S1 Table of S1 Appendix.

**3.3.2. Independent variable.** The prior literature proved that MWP was a function of risk, resilience, environment, and development. We would select the independent variables from the above four perspectives:

1. Risk. Disease was the most important risk affecting MW, as the main reason people used countless PPEs was to combat the disease. Accordingly, we chose Disease Mortality (DM) as an indicator to assess risk. It was defined as the number of deaths due to diseases (including cancer, diabetes, heart disease, AIDS, etc.) per 1,000 population (Deaths/1K Population). DW was derived from Center for Disease Control and prevention (CDC), available at (https://www.cdc.gov/datastatistics/index.html), which was a continuous variable with values ranging from 0.32 to 657.62 and a full sample mean of 134.28.

2. Resilience. We used the Medical Index (MI) to assess the resilience for a given country, derived from the sum of two indicators, the Healthcare Access and Quality index (HAQ) and Current Health Expenditure (CHE). The HAQ was obtained from a representative

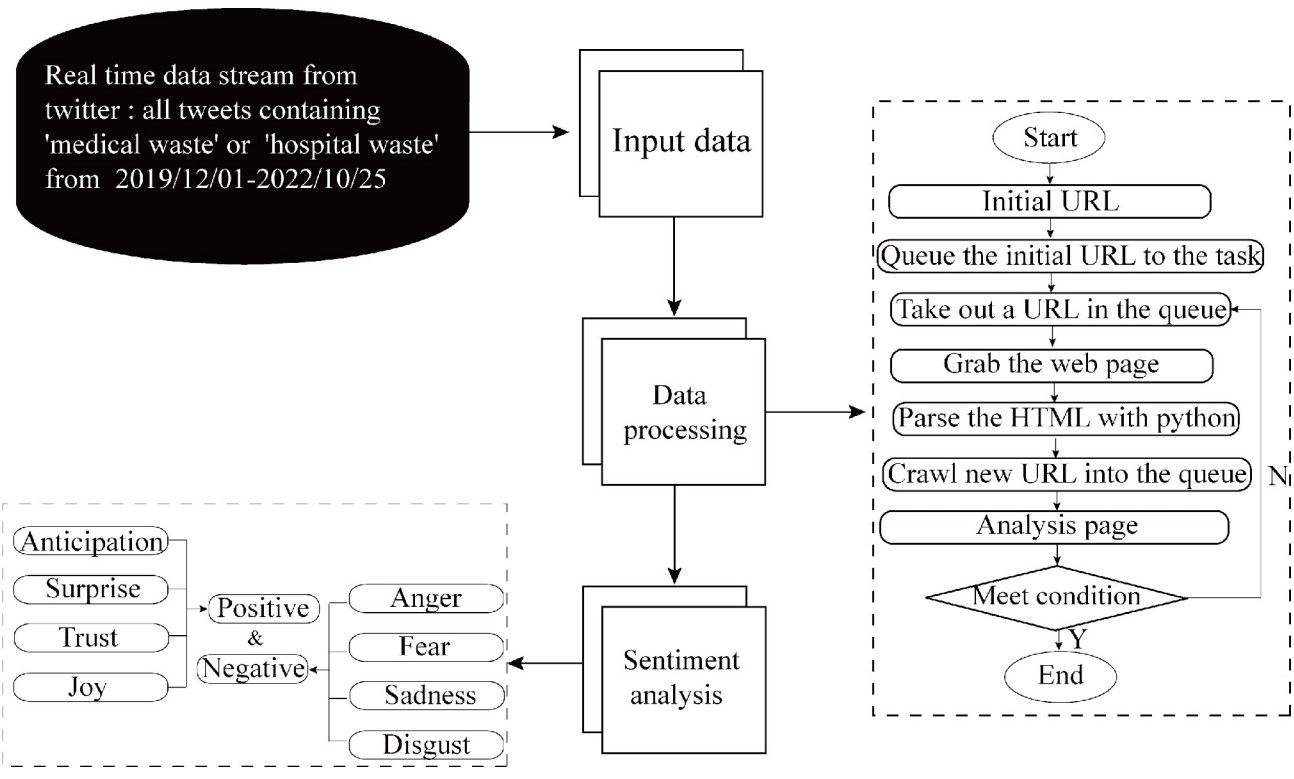

**Fig 1. Data acquisition process.**

study by Fullman, recorded the accessibility and quality of healthcare resources in 195 countries worldwide, available at (https://www.sciencedirect.com/science/article/pii/S0140673618309942). The CHE, similarly, was the publicly available data from the World Bank 2022 financial reports, denoted the value of health expenditure as a percentage of total GDP, which available at (https://data.worldbank.org/indicator/SH.XPD.CHEX.GD.ZS). HAQ and CHE were continuous variables with values ranging from 19–97 and 1.80–16.77. We quintile HAQ and CHE to generate the MI, which taking values as: very low = 1, low = 2, medium = 3, high = 4, and very high = 5. The mean value of MI in the full sample was 2.02.

3. Environment. We used the Environmental Performance Index (EPI) to characterize the environmental conditions of a given country for the public participation in MW. The 2022 Yale university Environmental Performance Index (EPI) reports provided a data-driven summary of the state of sustainability around the world, which used 40 performance indicators across 11 issue categories, and ranked 200 countries on climate change performance, environmental health, and ecosystem vitality. EPI had been shown by academic researchers to be a key variable in characterizing environmental sustainability. EPI data can be obtained through (https://epi.yale.edu/downloads), which was a continuous variable with a value range of 18.9–77.9 and a full sample mean of 43.84.

4. Development. We constructed a comprehensive indicator, the Social Development Index (SDI), to capture the development of a given country. The SDI covered three sub-indicators: Human Development Index (HDI), Life Expectancy at Birth (LEB), and Expected Years of Schooling (EYS). These three indicators were chosen because they were proved to

be well suited in assessing human well-being. HDI, LEB, EYS were derived from the human development report of the United Nations Development Programme, available at (https://hdr.undp.org/system/files/documents//hdr2020pdf). HDI, LEB, and EYS were continuous variables with values ranging from 0.394–0.962, 52.5–84.8, and 7.4–21.1, respectively. To generate the SDI, we quintile these three sub-indicators, taking values in the range of very low = 1, low = 2, medium = 3, high = 4, and very high = 5, with a mean value of 2.02.

**3.3.3. Control variable.** Considering that our data were cross-country, too many control variables might lead to potential statistical bias. However, we believed that Internet Access Rate (IAR) was still needed to be controlled for, as it led to digital divide, shaped frequency heterogeneity in the public use of social media and ultimately biased the results. The IAR referred to the internet access rate for a given country, and was the population with internet access divided by the total population. IAR was from the World Bank's 2022 Internet Report, available at (https://ourworldindata.org/internet), which was a continuous variable with a value range of 9.4–100 and a sample mean of 64.96.

The conceptualization and operationalization of all variables were shown in Table 1.

**3.3.4. Spatial regression models.** To estimate the mechanism influenced cross-country heterogeneity of MWP, we constructed the spatial regression model (SRM). The methodological reason for choosing the spatial regression model over other models was that the SRM is well suited to deal with estimation bias due to spatial autocorrelation caused by the geographic location of individuals, especially in cross-country analysis. The first step was to verify that MWP were spatially auto-correlated, as follows:

$$\begin{cases} I_{i1} = \dfrac{\sum_{i=1}^{n} \sum_{j=1}^{n} w_{ij}(x_i - \bar{x})\left(x_j - \bar{x}\right)}{S^2 \sum_{i=1}^{n} \sum_{j=1}^{n} w_{ij}} \\[2em] I_{j1} = \dfrac{(x_i - \bar{x})}{S^2} \sum_{j=1}^{n} w_{ij} \end{cases} \tag{1}$$

Where the upper part of Eq (1) ($I_{i1}$) was the global Moran index and the lower part ($I_{i2}$) was the local Moran index, explored spatial auto-correlation from the global and the local, respectively. x denoted MWP, i and j denoted the spatial weight matrix constructed according to the given country longitude, dimension, $w_{i,j}$ denoted the spatial weight matrix to measure the spatial distance between region i and region j. $S^2$ was the sample variance. The two sets of Moran index shared the same range of value from -1 to 1. A negative value indicated the existence of negative spatial correlation, a positive value indicated the positive, and a value closed to 0 indicated a more random spatial distribution.

After conducted the Moran index test, we next performed two spatial regression models. The first was the Spatial Lagged Model (SLM), as shown in Eq (2):

$$y = \lambda\omega_y + X\beta + \varepsilon \tag{2}$$

where X was the matrix of independent variables, indicated the spatial distribution of DM, MI, EPI, SDI in 141 countries, and y was the matrix of dependent variables, indicated the spatial distribution of WSP. $\omega_y$ denoted the matrix of spatial weights, $\lambda$ denoted spatial autoregressive coefficients, and $\beta$ was the matrix of parameters to be estimated, namely the random disturbance terms.

The second model was the Spatial Error Model (SEM), as shown in euqation (3):

$$y = X\beta + \rho\omega_{\mu} + \varepsilon \tag{3}$$

**Table 1. Conceptualization and operationalization of dependent variables (MWP), independent variables (DM, MI, EPI, SDI) and covariates (IAR).**

| Variables | Conceptualization | Operationalization |
|---|---|---|
| MWP | whether the public attitude toward MW in a given country was positive or negative. | Derived from crawler and sentiment analysis.[a] |
| Risk(DM) | The number of deaths due to major diseases per 100,000 population. | DM |
| Resilience(MI) | The level of medical healthcare for a given country. | (HAQ[b]+CHE[c])/2 |
| Environment(EPI) | The summary of the environmental sustainability for a give country. | EPI |
| Development(SDI) | The level of social development for a given country. | (HDI[d]+LEB[e]+EYS[f])/3 |
| IAR | Internet Access Rate for a given country. | IAR |

a. Weighted sentiment score (0 to 100, higher is more positive) for tweets from a given country.

b. Healthcare Access and Quality index, which evaluated the level of medical care in a given country.

c. Current Health Expenditure, which evaluated the level of investment in healthcare in a given country.

d. Human Development Index, which evaluated the level of social development in a given country.

e. Life Expectancy at Birth, average of life expectancy of the population in a given country.

f. Expected Years of Schooling, Average of the expected years of schooling of the population in a given country.

where X was the matrix of independent variables, indicated the spatial distribution of DM, MI, EPI, SDI, and y denoted the matrix of MWP. The β was the matrix of parameters to be estimated. And $\omega_\mu$ denoted the spatial distance matrix, μ was the spatial error term, and ε was the regression error term.

The analysis controled for MW risk, environment, development, resilience and IAR in each country, and used a spatial regression model that geographically weights all the potential variables, accordingly it can be assumed that the sample of selected countries was inherent unbiased. This study was approved by the Ethics Review Committee of Xi'an Jiaotong University (ID:2023011622).

## Section 4: Result

### 4.1. Descriptive statistics

Descriptive statistics for all sub-indicators in 141 specific countries were systematically shown in Fig 2 (The deeper the color was the higher the value).

The geographic distribution of the selected variables was as fellow: The DM, MI, SDI, IAR were all higher in Europe, followed by South America, both of which were developed countries. The distribution in Africa was more complex, with some countries in North Africa (relatively developed) shared higher and those in South Africa (relatively lagging) suffering from the lower. Taken together, the advantaged regions shared higher positive public attitudes of MW.

Fig 3 captured the temporal evolution of public concern for MW from 2019/12 to 2022/12. A striking feature was that public concern for MW was much higher during the COVID period (2019/12-2021/4) than in other periods. This distribution was due to the fact that the public used a large number of PPEs including discarded masks, test papers, syringes, etc. during the New Crown period. Furthermore, the global population paid more attention to life and health during this period because of the high lethality and prevalence of COVID-19.

### 4.2. Potential mechanisms shaping cross-country differences in MWP

**4.2.1. Basic spatial regression estimation.** The basic spatial regression model was shown in Table 2, which revealed the influence of selected variables on MWP. Model 1 was the Ordinary Least Squares (OLS) and model 2 based on the SLM. In line with the AIC and BIC criteria, we chose model 2 to report the results. We captured that MI(β = 0.78, SD = 0.14, P<0.01)

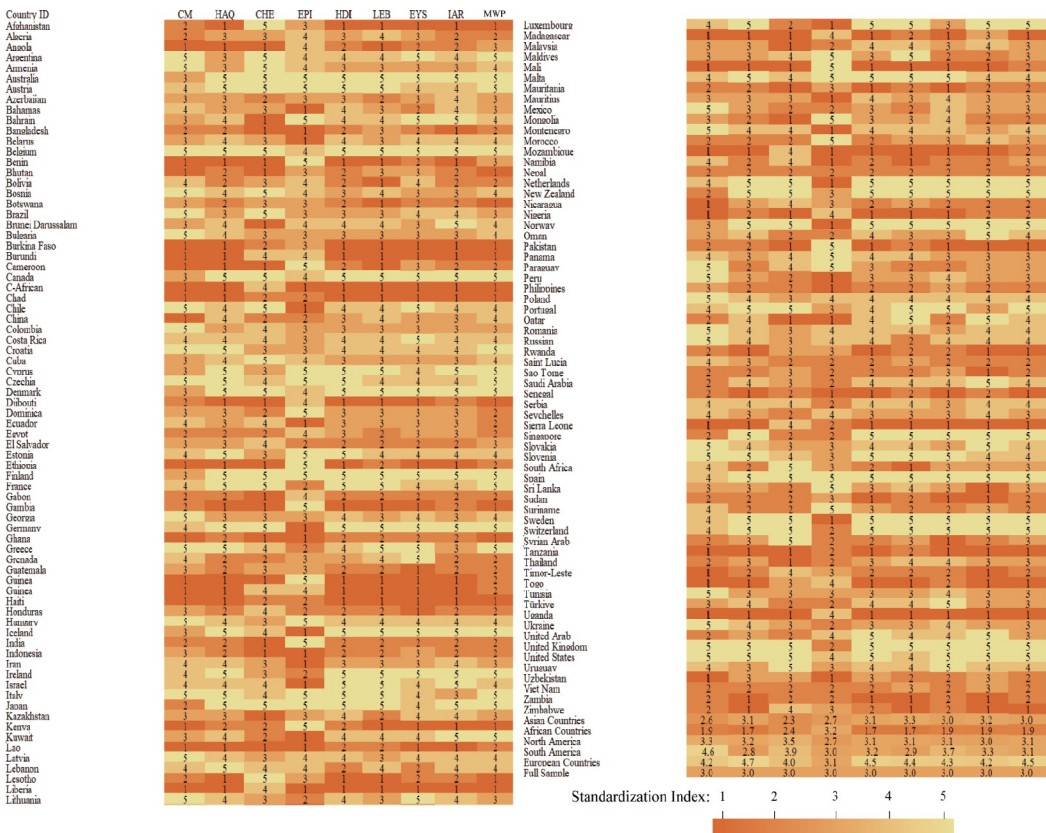

**Fig 2. Data description.**

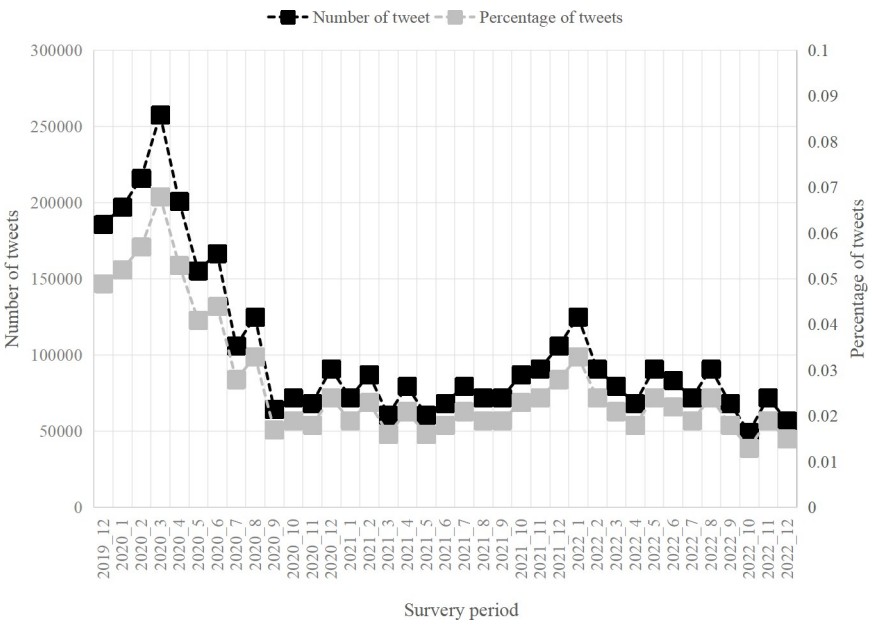

**Fig 3. Temporal evolution of public concern for MW from 2019/12 to 2022/12.**

and SDI (β = 1.66, SD = 0.13, P<0.01) posed a statistically significant positive influence on MWP, while DM (β = -0.1, SD = 0.12, P>0.05) and EPI (β = 0.09, SD = 0.09, P>0.05) were not significant.

**4.2.2. Moderated spatial regression estimation.** The present study continued to test the potential moderating inlfluence of selected variables on MWP, the results were presented in Table 3. Models 3, 4, 5 were the moderation of DM with MI, EPI, SDI, respectively, and the results were positive (C = -0.17, SD = 0.08), negative (C = -0.07, SD = 0.03), positive (C = 0.05, SD = 0.02), statistically significant simultaneously. The above scenario suggested that DM amplified the positive influence of MI and SDI on MWP. Meanwhile, models 7 indicated that MI positively moderated the influence of SDI (C = 0.29, SD = 0.07) on MWP. Taken together, there were significant moderation between selected variables on MWP.

**4.2.3. Heterogeneity analysis.** After analyzing the average influence, we then focused on the heterogeneity of the influence in different regions to capture more abundant and targeted conclusions. Models 9–14 systematically presented the subgroup analysis for different regions, as shown in Table 4. Firstly, the results of Model 9 and 12 indicated that the influence of Asian and North-America remained largely consistent with the full sample. Secondly, Africa (Model 11) and South America (Model 13) shared the same trend as follows: DM was more positively significant, SDI was smaller, and MI was insignificant, compared to the full sample, it is notable that both of these are extremely lagging regions. Thirdly, in Europe, the most developed countries worldwide, the influence of DM, MI, EPI, SDI were lower. Taken together, the present heterogeneity analysis revealed the following scenario: MWP in poor regions would be more sensitive to DM because of the unfavorable MI and SDI, while, developed regions had already shaped the capacity to resist MW risks as resilience, development and the environment.

**4.2.4. Robustness.** To verify the robustness, we replace the sentimental lexicon, using Sentiwordnet lexicon obtained from (https://github.com/aesuli/SentiWordNet.) and NTUSD lexicon obtained from (http://nlg.csie.ntu.edu.tw/nlpresource/NTUSD-Fin/.) Additionally, we replaced the geographic distance matrix in the spatial regressions with the economic distance matrix. The economic distance denoted the frequency and intensity of economic interactions

**Table 2. Basic predictive model for cross-country heterogeneity of MWP.**

| Variables | Model1[a] | | Model2[b] | |
|---|---|---|---|---|
| | β[c] | SD | β[c] | SD |
| DM | -0.04 | 0.12 | -0.1 | 0.12 |
| MI | 0.80** | 0.43 | 0.78*** | 0.14 |
| EPI | 0.03 | 0.08 | 0.09 | 0.09 |
| SDI | 1.55*** | 0.13 | 1.66*** | 0.13 |
| AIC[d] | 788.23 | | 433.56 | |
| BIC[e] | 617.92 | | 388.98 | |
| $\sqrt[2]{AIC^2 + BIC^2}$ | 1001.56 | | 582.48 | |
| CONS | 60.23*** | 0.42 | 62.33*** | 0.68 |
| R$^2$ | 0.8355 | 0.8348 | 0.8505 | 0.8395 |

a. A liner model with OLS.

b. A liner model with SLM.

c. *, **, *** represent significant at 10, 5, 1 percentage.

d. Akaike information criterion, used to evaluate the model fitting.

e. Bayesian information criterion, used to evaluate the model fitting.

**Table 3. Moderated predictive model for cross-country heterogeneity of MW[a].**

| Variables | Model 3[b] | Model 4[b] | Model 5[b] | Model 6[b] | Model 7[b] | Model 8[b] |
|---|---|---|---|---|---|---|
| DM | -0.68*** (0.28) | 0.12 (0.23) | -0.25 (0.25) | -0.1 (0.12) | <0.01 (0.12) | -0.11 (0.12) |
| MI | 0.32 (0.25) | 0.80*** (0.14) | 0.77*** (0.14) | 1.01*** (0.23) | -0.07 (0.24) | 0.83*** (0.14) |
| EPI | 0.06 (0.09) | 0.3 (0.21) | 0.08 (0.09) | 0.31 (0.20) | 0.03 (0.08) | 0.51*** (0.20) |
| SDI | 1.66*** (0.13) | 1.65*** (0.13) | 1.52*** (0.24) | 1.66*** (0.13) | 0.60*** (0.27) | 2.04*** (0.21) |
| DM*MI[c] | 0.17*** (0.08) | - | - | - | - | - |
| DM*EPI[d] | - | -0.07*** (0.03) | - | - | - | - |
| DM*SDI[e] | - | - | 0.05*** (0.02) | - | - | - |
| MI*EPI[f] | - | - | - | -0.07 (0.06) | - | - |
| MI*SDI[g] | - | - | - | - | 0.29*** (0.07) | - |
| EPI*SDI[h] | - | - | - | - | - | -0.17(0.06) |
| CONS | 61.62*** (0.74) | 59.26*** (0.74) | 60.60*** (0.68) | 59.49*** (0.74) | 62.71*** (0.69) | 58.89*** (0.71) |
| $R^2$ | 0.8404 | 0.8358 | 0.8348 | 0.8361 | 0.8547 | 0.8407 |

a. All model used SLM.

b. *, **, *** represent significant at 10, 5, 1 percentage.

c. Generate the interaction term between DM and MI (multiply the two).

d. Generate the interaction term between DM and EPI (multiply the two).

e. Generate the interaction term between DM and SDI (multiply the two).

f. Generate the interaction term between MI and EPI (multiply the two).

g. Generate the interaction term between MI and SDI (multiply the two).

h. Generate the interaction term between EPI and SDI (multiply the two).

across the selected countries, which from James and Chin's representative work [37]. The results were shown systematically in Table 5. The results revealed that the robustness was maintained after replacing the lexicon and spatial matrix.

**Table 4. Heterogeneity model[a].**

| Variable | Model 9[b] | Model 10[b] | Model 11[b] | Model 12[b] | Model 13[b] | Model 14[b] |
|---|---|---|---|---|---|---|
| DM | -0.1 (0.12) | 0.19 (0.27) | 0.54*** (0.25) | -0.72 (0.49) | 1.39*** (0.50) | -0.36 (0.26) |
| MI | 0.78*** (0.14) | 0.85*** (0.23) | 0.12 (0.23) | 1.54*** (0.55) | 0.32 (0.41) | 0.46 (0.55) |
| EPI | 0.09 (0.09) | 0.06 (0.17) | 0.17 (0.14) | -0.60 (0.45) | 0.27 (0.17) | -0.13 (0.13) |
| SDI | 1.66*** (0.03) | 1.44*** (0.24) | 1.06*** (0.28) | 1.70*** (0.56) | 1.41*** (0.35) | 1.64*** (0.38) |
| CONS | 60.23*** (0.42) | 60.10*** (0.96) | 61.10*** (0.65) | 60.41*** (1.98) | 54.63*** (1.27) | 64.23*** (2.69) |
| $R^2$ | 0.8355 | 0.7235 | 0.6646 | 0.7655 | 0.8971 | 0.6243 |
| Region | Full[c] | Asian[d] | African[e] | North-A[f] | South-A[g] | European[h] |

a. All model used SLM.

b. *, **, *** represent significant at 10, 5, 1 percentage.

c. Full sample included all 141 countries.

d. Asian sample included 32 Asian countries.

e. African sample included 38 African countries.

f. North-America sample included 19 North-America.

g. South-America sample included 22 South-America.

h. European sample included 30 European countries.

## Section 5: Discussion

Nowadays, MW is posing massive threat to public health for all human-beings [1, 4]. The present study revealed the public attitudes towards MW from a global perspective, meanwhile, discuss the potential determinants related to MW public attitudes. The following meaningful conclusions were captured:

1. Public attitudes towards MW were generally positive, with significant cross-national differences. We collected 3,789,764 tweets from 141 countries, 64.77% individuals in 45 Africa countries, 71.92% in 36 Europe countries, 69.59% in 60 Asia, South America and North America countries posed positive attitudes to MW. Overall, in the full sample over 60% shared positive attitudes towards MW which significantly exceeded other environmental behaviors such as waste separation, recycled water, and the greenhouse [38]. Additionally, the developed regions shared the higher MWP, compared with their developing and lagging counterparts. The explanations of the above national disparities could be attributed to the fact that developed regions were equipped with more advanced MW disposal equipments, and therefore the public perceived lower MW risks in their daily lives [1, 2] Taken together, global stakeholders had initial successes in addressing MW risks, but significant international disparities remained.

2. Development (SDI) posed positive influence on public attitudes towards MW. We revealed that SDI ($\beta$ = 1.66, SD = 0.13, P < 0.01) posed a positive influence on MWP, which revealed the scenario that the more developed countries shared the more positive public attitudes of MW compared with their less counterparts. Extensive research has proved that developed countries shared more scientific MW degradation equipments and accordingly produced less MW-induced pollution [12, 23], while our study revealed the advantage of developed countries over developing or backward countries in terms of positive attitudes towards MW. The above academic scenario showed greater international inequality in MW, as developed countries with less MW pollution shared positive public attitudes towards MW, while developing and lagging countries with more MW pollution posed negative public attitudes. Additionally, the concept of SDI was first introduced and applied to public pro-environmental attitudes, the public with better social development tended to shaping more rational attitudes towards MW, and equipped with advanced MW treatment facilities. The above conclusion captured the necessity of promoting resilience and social progress in addressing MW Risks [1, 17].

**Table 5. Robustness test.**

| Variables | Model 15[a] | Model 16[a] | Model 17[a] | Model 18[a] |
|---|---|---|---|---|
| DM | 0.08(0.14) | -0.02 (0.04) | 0.14 (0.12) | 0.22(0.17) |
| MI | 1.14***(0.52) | 1.39***(0.65) | 1.73***(0.58) | 2.24***(1.03) |
| EPI | 0.03(0.03) | 0.32 (0.29) | 0.42 (0.39) | 0.57 (0.62) |
| SDI | 1.39***(0.44) | 1.87***(0.53) | 2.23***(0.77) | 2.65***(0.83) |
| Lexicon | Sentiwordnet[b] | NTUSD[c] | Sentiwordnet[b] | NTUSD[c] |
| Spatial weights | Geographic[d] | Geographic[e] | Economic[e] | Economic[e] |

a. *, **, *** represent significant at 10, 5, 1 percentage.

b. Another Sentiment Analysis Dictionary.

c. Another Sentiment Analysis Dictionary.

d. Geographic distance matrix.

e. Intensity of economic interactions across the selected countries.

3. Resilience (HAQ) positively influenced the public attitudes towards MW. The present empirical study indicated that both MI (β = 0.78, SD = 0.14, P < 0.01) posed a positive and significant influence on MWP. Resilience was a widely used indicator in the social sciences to characterize the ability to withstand risk, the present result suggested that improving healthcare accessibility and quality would guarantee that residents have more confidence in MW management, which promoted to alleviate public anxiety [29, 30, 32].

4. Risk (DM) and environment (EPI) were irrelated to public attitudes towards MW directly. The results revealed that the influence of DM (β = -0.1, SD = 0.12, P > 0.05) and EPI (β = 0.09, SD = 0.09, P > 0.05) on MWP was statistically insignificant. An explanation of the above scenario was that the public was excluded from MW management for a long time, shaped them unaware of the potential risks and environment, as we summarized in the literature review. The above phenomenon was dangerous, especially in the context of multi-use medical devices were turn into single-use medical devices and bring about innumerable household PPEs in the COVID-19 [19, 20]. Of particular interest was that there were indirect moderation of risk and environment on MWP. Previous academic research integrated risk and environment as environmental perception or environmental risk perception and positively influenced pro-environmental behavior [5, 6, 23]. Our findings were inconsistent with previous analyses of pro-environmental behaviors due to the long-term exclusion of the public from MW management, leaving them unaware of MW risks.

5. There were positive moderating influences between selected factors on MWP. The present study revealed that risk (CM) positively moderated the influence of resilience (MI) and development (SDI) on MWP, and resilience (MI) positively moderated the influence of development (SDI) on MWP. The above scenario implied that the positive influence of resilience and development on the MWP was stronger in regions with higher risk, suggesting that risk post indirect influence on the MWP. The positive moderation between resilience and development was not captured in the established studies, and accordingly meaningful, as developed regions always shared more resilience [2, 22]. Taken together, the above conclusion captured that the inter-regional disparities of public attitudes toward MW would be widened in the foreseeable future, due to that advantaged conditions or resources were solely in the hands of a few developed countries, than their developing or lagging counterparts [17, 19].

6. There were cross-regional disparities in the influence of the selected factors on the MWP. More specifically, public attitudes towards MW in extremely lagging regions were more likely to suffered from the disadvantage in risks (DM), resilience (MI) and development (SDI). The heterogeneous results indicated that Africa and South-America (extremely lagging regions) shared the trend as follows: DM influence was more positively significant, SDI influence was smaller, and MI influence was insignificant, compared to the full sample. On the one hand, this implied that the influence of SDI and MI on MWP were lower in poor areas, simultaneously, because they hardly equipped with advanced medical treatments and knowledge [4–6]. This reminded policymakers of the need to strengthen development and resilience in lagging areas, and protect them from the risk of global epidemic [8, 23, 26].

Taken together, we obtained the following complex scenario: Overall, the global public attitudes towards MW were positive, with significant cross-national disparities. However, it's pessimistic that those MWP determinants (resilience and development) tended to be controlled solely in developed countries, which indicated that the cross-nation disparities would be widen in the future. Several established researches captured that lagging countries was suffering from severe MW risk such as low technology, insufficient scientific knowledge, and

disadvantaged economic foundation [36, 37]. Especially during the 21st century, the incineration plants used for MW treatment were often equipped with old and unsafe technologies in undeveloped countries, which caused further heavy solid pollution and dioxin emissions, the emission rate of these toxins could go as high as 40,000 times higher than the Stockholm Convention's emission limits, and seriously harmed the health of vulnerable residents [1, 4].

Combining the obtained conclusions, we propose the following three policy recommendations:

1. The government should mobilize public participation in MW management. From this perspective, three realistic approaches are for stakeholders to consider. Firstly, the government needs to set up a special agency to collect public opinions on MW, for example, the America government integrated the three organizations, to set a specific MW management committee [6]. Secondly, the government should organize professional staff to provide professional MW knowledge to the public, for instance, France, the UK and Canada all shared the specialised scientists in MW management [12].

2. Countries should improve their resilience to deal with the MW risks. From this perspective, improving the quality and accessibility of healthcare is essential. Stakeholders should set standardized MW governance regulations, just like the standardized regulations they developed on MW emissions, international organizations should develop standardized MW management regulations and aiming to reducing MW inequalities across countries, pay special attention to public particiaption.

3. Proper segregation of MW in healthcare facilities is required. The stakeholders should action as recommended by WHO, applying more scientific MW disposal, like chemical disinfection and steam sterilisation. They were both heat-based, safe and efficient treatment processes and were the most popular methods of MW disposal. If segregation is difficult and cannot be separated from general waste, the hospitals must set up stations to treat the MW and reduce secondary infections caused by recycling [38].

In addition, this paper has some shortcomings: Firstly, from a methodological perspective, only the endogeneity due to spatial auto-correlation is balanced, but not the endogeneity from other statistical bias. Spatial difference-in-difference model is a good choice, but many data are missing, making it impossible to use this model. Secondly, it is difficult to uncover the potential mechanisms that shaping the heterogeneity in MWP across-country. Thirdly, randomized controlled trials (RCTs) was not taken in the present study. Traditional crawler-based sentiment analyses are difficult to incorporate into RCTs, which also makes it difficult to independently isolate the net link between dependent and independent variables. Although we have used geographically weighted regressions and come to more accurate conclusions, there is still a gap with the requirements of an RCTs.

Despite these shortcomings, we still provide a relatively complete analytical framework for cross-country experience of MW management, and draw some meaningful conclusions. Accordingly, we have the following implications for future research: (1) more cross-country analysis based on big data needs to be applied to other areas of environmental science, (2) revealing the potential mechanisms shaping MWP differences and (3) MW reviews and historical studies are needed in the future.

## Section 6: Conclusion

The present study analyzed the public attitudes towards MW and revealed it's cross-country heterogeneity, as well as the potential determinants. Public attitudes towards MW were

generally positive, but there were large cross-national disparities. Stakeholders would need to designate targeted strategies to enhance public satisfaction with MW management.

## Supporting information

**S1 Appendix.**
(DOCX)

## Acknowledgments

All the authors want to thank the staff in Southwest University of Finance and Economics. We are also grateful to all college students who took part in this study.

## Author Contributions

**Conceptualization:** Zhipeng Bai.

**Data curation:** Zhipeng Bai, Xi Liu.

**Formal analysis:** Zhipeng Bai, Xi Liu.

**Funding acquisition:** Zhipeng Bai, Xi Liu.

**Investigation:** Xi Liu.

**Methodology:** Zhipeng Bai, Xi Liu.

**Project administration:** Zhipeng Bai, Xi Liu.

**Resources:** Zhipeng Bai, Xi Liu.

**Software:** Zhipeng Bai, Xi Liu.

**Supervision:** Xi Liu, Wenbao Ma.

**Validation:** Wenbao Ma.

**Visualization:** Wenbao Ma.

**Writing – original draft:** Wenbao Ma.

**Writing – review & editing:** Wenbao Ma.

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
