## [Decision Letter · Decision Letter 0]

22 Oct 2023

PONE-D-23-22632Public attitudes toward medical waste: Experiences from 141 countries.PLOS ONE

Dear Dr. Bai,

Thank you for submitting your manuscript to PLOS ONE. After careful consideration, we feel that it has merit but does not fully meet PLOS ONE’s publication criteria as it currently stands. Therefore, we invite you to submit a revised version of the manuscript that addresses the points raised during the review process.

We look forward to receiving your revised manuscript.

Kind regards,

Meghana Ray, Ph.D., MBA, B.Pharm

Academic Editor

PLOS ONE

2. In your Methods section, please include additional information about your dataset and ensure that you have included a statement specifying whether the collection and analysis method complied with the terms and conditions for the source of the data.

5. PLOS requires an ORCID iD for the corresponding author in Editorial Manager on papers submitted after December 6th, 2016. Please ensure that you have an ORCID iD and that it is validated in Editorial Manager. To do this, go to ‘Update my Information’ (in the upper left-hand corner of the main menu), and click on the Fetch/Validate link next to the ORCID field. This will take you to the ORCID site and allow you to create a new iD or authenticate a pre-existing iD in Editorial Manager. Please see the following video for instructions on linking an ORCID iD to your Editorial Manager account: "" ext-link-type="uri" xlink:type="simple">https://www.youtube.com/watch?v=_xcclfuvtxQ"".

7. We note that Figure 3 and 4 in your submission contain [map/satellite] images which may be copyrighted. All PLOS content is published under the Creative Commons Attribution License (CC BY 4.0), which means that the manuscript, images, and Supporting Information files will be freely available online, and any third party is permitted to access, download, copy, distribute, and use these materials in any way, even commercially, with proper attribution. For these reasons, we cannot publish previously copyrighted maps or satellite images created using proprietary data, such as Google software (Google Maps, Street View, and Earth). For more information, see our copyright guidelines: http://journals.plos.org/plosone/s/licenses-and-copyright.

a. You may seek permission from the original copyright holder of Figure 3 and 4 to publish the content specifically under the CC BY 4.0 license.  

Reviewers' comments:

Reviewer's Responses to Questions

**Comments to the Author**

1. Is the manuscript technically sound, and do the data support the conclusions?

Reviewer #1: Yes

Reviewer #2: Yes

2. Has the statistical analysis been performed appropriately and rigorously? 

Reviewer #1: Yes

Reviewer #2: N/A

3. Have the authors made all data underlying the findings in their manuscript fully available?

Reviewer #1: Yes

Reviewer #2: Yes

4. Is the manuscript presented in an intelligible fashion and written in standard English?

Reviewer #1: Yes

Reviewer #2: Yes

5. Review Comments to the Author

Reviewer #1: The paper is a very interesting study and a timely research work, analysing the public attitudes towards medical waste and revealing its cross-country heterogeneity and potential shaping mechanisms. It is a complete work; well written and structured with extensive literature review and comprehensive analyses. Excellent contribution to the body of knowledge in the respective field.

Reviewer #2: PLOS Review

"Public Attitudes Toward Medical Waste: Experiences from 141 Countries"

This mss. explores a needed topic in environmental health. The manuscript seems fairly good in its approach to this unexplored topic but may become crucial as we better understand public attitudes toward a series of environmental issues.

A question at the very beginning must be how do the major findings relate to how people in more developed nations priorly know about such issues to distinguish them from elsewhere ? It is intuitive that such advanced nations will have folks who are more aware of environmental issues. It is an interesting finding of resilience that can be meaningful for more strategizing for greater social action across nations.

Since this is an unfunded study, it reveals that this was significant and commendable work to assemble, retrieve, and analyze the massive data and then write the manuscript. However, a major question: Of those surveyed, what each person's awareness of medical waste and the dangers it may pose what was known beforehand? Was the study intended as an intervention to promote awareness? Is there any indication that this indeed happen?

Where does medical waste end up in various nations? Can the authors address this from the literature and popular press to query whether it affects public attitudes? The research also did not identify correlates to other pollution causes and can this receive some comment?

The research objective number 3 is absolutely key in terms of factors that influence public attitudes. It is noted that the time of the study was during the COVID outbreaks across the world so how might this epidemic heightened the findings in terms of generalized anxiety and concerns about population health? This could bear importantly upon the findings and the conclusions.

Medical waste management differs from community to community and surely differs within national borders. Also of great difference are the determinants of medical waste. Here in the manuscript there were several good observations.

As to methodology, this section in particularly needs a little more clarity and shortening. Twitter and crawler mining is an interesting and inexpensive mechanism for research into public views, but what inherent biases might come from these accounts where a large database is needed? Should there be a list of all the nations surveyed and perhaps percentages of the 1.3 million responses, which is somewhat amazing. The tables could receive a little more explanatory commentary for readers to understand better.

Finally, English editing is most important where some of the areas are rather tedious to read, but overall it is an important study that revealed cross-country heterogenicity and the potential shaping mechanisms.

It is key to continue to examine global public attitudes toward medical waste and numerous other environmental issues. How public attitudes are influenced and begin to change is important to map, as well as probe the reasons why people perceive certain environmental risks.

In the ending paragraphs, the policy recommendations appear rather broad and could use a bit more specificity for practical responses by policymakers.

Sincerely

J.Warrren Salmon, Ph.D.

P.S. This reviewer will be interested in reviewing the next submission of this manuscript.

6. PLOS authors have the option to publish the peer review history of their article (what does this mean?). If published, this will include your full peer review and any attached files.

Reviewer #1: **Yes: **Hasim Altan

Reviewer #2: No

---

## [Author Response · Author response to Decision Letter 0]

11 Mar 2024

Dear Reviewer:

I would like to thank you for your detailed revisions of this manuscript, which have improved the quality of this manuscript in many ways and have helped me to advance my academic career and that of my collaborators, for which I am grateful! I have responded to your comments on a one-to-one document as follows:

Question 1:

A question at the very beginning must be how do the major findings relate to how people in more developed nations priorly know about such issues to distinguish them from elsewhere ? It is intuitive that such advanced nations will have folks who are more aware of environmental issues. It is an interesting finding of resilience that can be meaningful for more strategizing for greater social action across nations. 

Answer 1:

Thank you for your meaningful review. We agree with you and believe that highlighting how people in more developed nations prior know about such issues to distinguish them from elsewhere is the focus of this study. We adopt the following approach: firstly, we emphasise the importance of revealing that MW public attitudes are more positive in developed countries, which tend to have higher MW emissions and higher amounts of pollution due to MW as shown in lines 51-55; secondly, we highlight the study's "Identifying whether there are significant differences in MW public attitudes between developed and developing countries" as the purpose of the study, as shown in lines 72-73; third, we focus on the relationship between the level of development of a given country and MW attitudes in the " Discussion" section, as shown in lines 345-358.

Question 2:

Since this is an unfunded study, it reveals that this was significant and commendable work to assemble, retrieve, and analyze the massive data and then write the manuscript. However, a major question: Of those surveyed, what each person's awareness of medical waste and the dangers it may pose what was known beforehand? Was the study intended as an intervention to promote awareness? Is there any indication that this indeed happen?

Answer 2:

Thank you for your meaningful suggestions. Of course, our paper aims to study how to improve public awareness of MW, and we also propose relevant solutions through our cross-sectional analyses, and our study is focusing on public attitudes towards MW. Of course, we understand the importance of randomised controlled trial for the experiment, but we can't obtain the awareness of medical waste that each person possessed before the survey. we just reveal that the public attitude towards MW is related to Risk, Resilience. Environment, and Development variables of interest. However, as we stated above, this is only a cross-sectional study, and we did not impose the exact intervention, and therefore cannot fully causally account for the effect of imposing a given intervention (e.g., improving development, or improving frustration tolerance) on public attitudes toward MW. Our conclusions can only be partially illustrated in the cross-section by analysing the underlying trends in the data based on statistical methods to illustrate how to enhance positive public attitudes towards MW. We clearly understand the importance of RCTs for the scientific validity of experiments, but studies based on sentiment analyses rarely use RCTs, and we have added geographically weighted regressions to traditional sentiment analyses, thus drawing more realistic conclusions. Of course, we would cite the lack of use of RCT as a limitation of this study, as shown in line 433-437.

Question 3:

Where does medical waste end up in various nations? Can the authors address this from the literature and popular press to query whether it affects public attitudes? The research also did not identify correlates to other pollution causes and can this receive some comment?

Answer 3:

Thank you for your meaningful review. We think that the issue you raise is important and we have added a large amount of literature on the heterogeneity of the way MW is disposed of in different countries, as shown in lines 120-130. But unfortunately, there is no official document or any literature that quantifies a specific indicator of the way MEDICAL WASTE ENDS UP in different countries, and we have gone through nearly fifty or so related papers, checking the ddocuments of Center for Disease Control and prevention (CDC), available at (https://www.cdc.gov/datastatistics/index.html），World Bank 2022 reports, available at (https://data.worldbank.org/indicator/), Environmental Performance Index (EPI) reports(https://epi.yale.edu/downloads), United Nations Development Programme, available at (https://hdr.undp.org/system/files/documents//hdr2020pdf), Unfortunately, however, no quantifiable indicators have been found on the way in which medical waste ends up in different countries.

On the other hand, as far as you are concerned, the association with other pollution causes is not the focus of this study. I make the following explanation; because the source of medical pollution is relatively homogeneous, the vast majority of it originates from hospitals and a small portion from households, and medical pollution is hospital-led or government-led in the process of dealing with it, which means that it is detached from the public. This unique characteristic determines that he is not as ubiquitous in everyone's daily life as other causes of pollution, such as air pollution, water pollution, light pollution. So we did not choose to control these causes of pollution. In addition, because of the large number of countries involved in our study, and the difficulty of finding high-quality data on the causes of pollutionof all valid countries, the inclusion of some of the variables could lead to too many missing values and affect the validity of the statistical conclusions.

Question 4:

The research objective number 3 is absolutely key in terms of factors that influence public attitudes. It is noted that the time of the study was during the COVID outbreaks across the world so how might this epidemic heightened the findings in terms of generalized anxiety and concerns about population health? This could bear importantly upon the findings and the conclusions.

Answer 4:

Thank you for your meaningful review. We think the question you raise is important, and we have added information about the temporal trends in public interest in MW during COVID-19 and as an important focus of this study. As shown in lines 279-284 and figure 3.

Question 5:

As to methodology, this section in particularly needs a little more clarity and shortening. Twitter and crawler mining is an interesting and inexpensive mechanism for research into public views, but what inherent biases might come from these accounts where a large database is needed? Should there be a list of all the nations surveyed and perhaps percentages of the 1.3 million responses, which is somewhat amazing. The tables could receive a little more explanatory commentary for readers to understand better.

Answer 5:

Thank you for your meaningful review. We think the issue you raise is important, and we have removed a large number of unimportant statements in the METHODOLOGY section, from 1,651 words to 1,234 words. Regarding the second issue, inherent biases in the selection of the country sample, we controlled for each country's exposure to MW risk, environment, level of development, resilience, and Internet connectivity in our analyses in the manuscript, as well as using a spatial regression model that geographically weighted all latent variables, and thus can be considered to select a sample of countries with essentially no inherent biases. We account for this phenomenon in the methodology section to ensure that we do not mislead the reader. Regarding the third question, we add an Appendix with Table S1 that meticulously illustrates the number of tweets about MW in all countries, sentiment attitudes to ensure the credibility of the study.

Question 6:

Finally, English editing is most important where some of the areas are rather tedious to read, but overall it is an important study that revealed cross-country heterogenicity and the potential shaping mechanisms.

Answer 6:

Thank you for your meaningful review. We invite our native English-speaking colleagues to carefully revise the manuscript in order to improve the level of English expression in this manuscript. As shown in the all “tracking” in the present manuscript.

Question 7:

In the ending paragraphs, the policy recommendations appear rather broad and could use a bit more specificity for practical responses by policymakers.

Answer 7:

Thank you for your meaningful review. We have rewritten the policy recommendations section to make it more practical. More specifically, we have cited experiences from developed countries such as the United States, Canada, and France, and introduced some relevant MW practices to make the policy recommendations more meaningful. As shown in lines 410-428.

---

## [Editor Report · Decision Letter 1]

5 Apr 2024

Public attitudes toward medical waste: Experiences from 141 countries.

PONE-D-23-22632R1

Dear Dr. Bai,

We’re pleased to inform you that your manuscript has been judged scientifically suitable for publication and will be formally accepted for publication once it meets all outstanding technical requirements.

Kind regards,

Meghana Ray, Ph.D., MBA, B.Pharm

Academic Editor

PLOS ONE

---

## [Editor Report · Acceptance letter]

29 Apr 2024

PONE-D-23-22632R1 

PLOS ONE

Dear Dr. Bai, 

I'm pleased to inform you that your manuscript has been deemed suitable for publication in PLOS ONE. Congratulations! Your manuscript is now being handed over to our production team.

Kind regards, 

on behalf of

Dr. Meghana Ray 

Academic Editor

PLOS ONE